# Inhibitory Potential of Synthetic Amino Acid Derivatives against Digestive Enzymes as Promising Hypoglycemic and Anti-Obesity Agents

**DOI:** 10.3390/biom13060953

**Published:** 2023-06-07

**Authors:** Franciane Campos da Silva, Bruna Celeida Silva Santos, Pedro Pôssa de Castro, Giovanni Wilson Amarante, Orlando Vieira de Sousa

**Affiliations:** 1Departamento de Ciências Farmacêuticas, Faculdade de Farmácia, Universidade Federal de Juiz de Fora, Campus Universitário, São Pedro, Juiz de Fora 36036-900, MG, Brazil; franciane.campos2010@gmail.com (F.C.d.S.); brunaceleida@gmail.com (B.C.S.S.); 2Departamento de Química, Instituto de Ciências Exatas, Universidade Federal de Juiz de Fora, Campus Universitário, São Pedro, Juiz de Fora 36036-900, MG, Brazil; pedro_possa@hotmail.com

**Keywords:** amino acid derivatives, digestive enzymes, pancreatic lipase, pancreatic α-amylase, α-glucosidase, metabolic disorders

## Abstract

Over the last decades, the increased incidence of metabolic disorders, such as type two diabetes and obesity, has motivated researchers to investigate new enzyme inhibitors. In this study, the inhibitory effects of synthetic amino acid derivatives (PPC80, PPC82, PPC84, PPC89, and PPC101) on the activity of digestive enzymes were assessed using in vitro assays. The inhibitory effect was determined by the inhibition percentage and the 50% inhibitory concentration (IC_50_), and the mechanism of action was investigated using kinetic parameters and Lineweaver–Burk plots. PPC80, PPC82, and PPC84 inhibited pancreatic lipase (IC_50_ of 167–1023 µM) via competitive or mixed mechanisms. The activity of pancreatic α-amylase was suppressed by PPC80, PPC82, PPC84, PPC89, and PPC101 (IC_50_ of 162–519 µM), which acted as competitive or mixed inhibitors. Finally, PPC84, PPC89, and PPC101 also showed potent inhibitory effects on α-glucosidase (IC_50_ of 51–353 µM) as competitive inhibitors. The results suggest that these synthetic amino acid derivatives have inhibitory potential against digestive enzymes and may be used as therapeutic agents to control metabolic disorders.

## 1. Introduction

The digestion of foods in the gastrointestinal tract of humans and animals is determined by the activity of enzymes that break down macronutrients into smaller molecules to be absorbed in the gut and used by the body [1]. Some of the most important digestive enzymes include α-amylase and α-glucosidase, which degrade carbohydrates to obtain energy, and lipases, which catalyze the cleavage of triglycerides to produce free fatty acids and monoacylglycerol that either meet metabolic needs or are re-esterified and stored as triglycerides in adipose tissue [2]. Diets rich in carbohydrates can lead to hyperglycemia, which is associated with high insulin levels in the blood and increased uptake of nutrients, leading to the accumulation of adipose tissue and obesity [3]. On the other hand, high-fat diets are associated with abnormally high levels of circulating fatty acids and subsequent ectopic deposition in non-adipose tissues as well as lipid accumulation in the liver, heart, endothelium, nervous system, pancreas, and skeletal muscle, thereby causing an imbalance in homeostatic mechanisms regulating metabolism [2,4]. This imbalance may lead to health complications such as metabolic disorders (e.g., dyslipidemia, hypertension, or type two diabetes), cancers, respiratory diseases, digestive problems, and osteoarthritis [5].

Regulation of nutrient absorption (e.g., carbohydrates) through the inhibition of digestive enzymes is an effective manner to control metabolism. For example, acarbose can inhibit the activity of α-amylase and α-glucosidase enzymes by reducing glucose absorption and decreasing insulin secretion in postprandial glycemia, establishing a glycemic control mechanism associated with reduced glycosylated hemoglobin. This class of enzymatic inhibitors is indicated in patients with adequate fasting blood glucose and elevated postprandial blood glucose levels. In patients with impaired glucose tolerance, enzymatic inhibitors have been associated with a marked reduction in cardiovascular events and no risk of adverse side effects, such as weight gain or hypoglycemia [6]. Therefore, the development of α-amylase and α-glucosidase inhibitors is increasingly recognized as a therapeutic strategy for patients with carbohydrate metabolic disorders, including postprandial hyperglycemia and type two diabetes mellitus [7,8].

Obesity is a complex disease that involves an abnormal or excessive accumulation of fat in the body and constitutes a public health problem worldwide [9]. The control and treatment of this pathology are mainly aimed at avoiding health complications as well as increasing life expectancy [9]. Among the available drugs, lipase inhibitors (e.g., orlistat) act by reducing the absorption of monoacylglycerol, thus leading to weight loss [7,8,10,11]. However, new synthetic anti-obesity agents, which may bring better benefits to patients, have been investigated [12].

In this context, the preparation of novel amino acid derivatives obtained from organic synthesis processes is a promising area that has been subjected to numerous biological studies. In addition to the functionalization of carboxylic and amine groups attached to the stereogenic center, the coupling of carbon side chains may also result in functional amino acid derivative drugs synthesized by conventional chemical reactions (i.e., acylation, alkylation, and amidation) [13]. These derivatives have attracted recent scientific interest due to their multiple biological properties [14,15]. For example, cationic antimicrobial peptides hold promise as new alternative antibiotics with the potential to inhibit multi-drug-resistant bacteria [16].

In the present study, the inhibitory effects of synthetic amino acid derivatives on digestive enzymes were assessed using in vitro assays. This exploratory study may have predictive value for developing new therapeutic agents against metabolic disorders such as type two diabetes mellitus and obesity.

## 2. Materials and Methods

### 2.1. Synthesis of Amino Acid Derivatives

The evaluated as amino acid derivatives, compounds PPC80 (342.52 g/mol), PPC82 (314.47 g/mol), PPC84 (287.40 g/mol), PPC89 (370.58 g/mol), and PPC101 (469.76 g/mol), were synthesized according to our previous report in the literature [17].

### 2.2. Chemicals

The drugs and reagents used in this study were as follows: porcine pancreatic lipase, 50 mmol/L Tris-HCl buffer (pH 8.0), *p*-nitrophenol palmitate, Triton-X 100, orlistat, porcine pancreatic α-amylase, 50 mmol/L Tris-HCl (pH 7.0), α-glucosidase, 100 mmol/L citrate-phosphate buffer (pH 7.0), acarbose, and *p*-nitrophenyl-α-*D*-glycopyranoside (Sigma-Aldrich^®^ Co., St. Louis, MO, USA), while dimethylsulfoxide and starch (Loja Synth^®^, Diadema, SP, Brazil). Unless noted, all chemicals utilized in the synthetic protocol were acquired from Sigma-Aldrich^®^ Co., St. Louis, MO, USA, and used as received.

### 2.3. Inhibitory Activity on Digestive Enzymes

#### 2.3.1. Pancreatic Lipase Inhibition Assay

The pancreatic lipase inhibition assay was performed according to Santos et al. [18] with some modifications. The porcine pancreatic lipase (10 g/L) was incubated in 50 mmol/L Tris-HCl buffer (pH 8.0) containing 10 mM CaCl_2_ and 25 mM NaCl. The *p*-nitrophenol palmitate substrate (8 mM) was dissolved in 0.5% *w*/*v* Triton-X 100. PPC80, while PPC82, PPC84, PPC89, and PPC101 amino acid derivatives and orlistat were solubilized in dimethylsulfoxide (DMSO) prepared at increasing concentrations ranging from 0.5 to 1.392 mM. A total of 100 µL of enzyme solution, 50 µL of *p*-nitrophenol palmitate substrate, and 50 µL of the amino acid derivative sample or orlistat were added to the microplate wells. Next, microplates were incubated at four different time intervals (10, 20, 30, and 40 min) in a water bath at 37 °C, and the reaction was stopped in an ice bath. All reactions were carried out in triplicate. The absorbance of the products was measured at 405 nm using a microplate reader (Thermoplate^®^, TP-Reader, Wuxi, China).

#### 2.3.2. Pancreatic α-Amylase Inhibition Assay

The pancreatic α-amylase inhibition assay was carried out according to Freitas et al. [19] with some modifications. The porcine pancreatic α-amylase (1 mg/mL) was incubated in 50 mM Tris-HCl buffer (pH 7.0) containing 10 mM CaCl_2_ and 1% starch. PPC80, PPC82, PPC84, PPC89, and PPC101 amino acid derivatives and acarbose were solubilized in DMSO prepared at increasing concentrations ranging from 0.15 to 1.590 mM. A total of 50 µL of enzyme solution, 50 µL of substrate, and 50 µL of amino acid derivative sample or acarbose were added to the microplate wells. Afterward, microplates were pre-incubated for 10 min in a water bath at 37 °C. A total of 100 µL of substrate was added to each well, and microplates were incubated at four different time intervals (10, 20, 30, and 40 min) in a water bath at 37 °C. The reaction was stopped using an ice bath. All reactions were carried out in triplicate. The absorbance of the products was measured at 405 nm using a microplate reader (Thermoplate^®^, TP-Reader, Wuxi, China).

#### 2.3.3. α-Glucosidase Inhibition Assay

The inhibitory effect against α-glucosidase was carried out according to Chelladurai and Chinnachamy [20] with some modifications. A total of 2 U/mL α-glucosidase and 5 mmol/L ρ-nitrophenyl-α-D-glucopyranoside substrate were solubilized in 100 mM citrate-phosphate buffer (pH 7.0). PPC80, PPC82, PPC84, PPC89, and PPC101 amino acid derivatives and acarbose were solubilized in DMSO prepared at increasing concentrations ranging from 0.24 to 1.740 mM. A total of 100 µL of α-glucosidase solution, 50 µL of amino acid derivative sample or acarbose, and 50 µL of substrate were added to the microplate wells. Afterward, microplates were incubated at different intervals (10, 20, 30, and 40 min) in a water bath at 37 °C. The reaction was stopped in an ice bath. All enzyme reactions were carried out in triplicate. The absorbance of the products was measured at 405 nm using a microplate reader (Thermoplate^®^, TP-Reader, Wuxi, China).

#### 2.3.4. Determination of the Inhibitory Effect and IC_50_

The percentage of inhibition (*I*%) was determined using “absorbance versus time” graphs. By means of linear regression, using the method of least-squares, the equations of the straight lines and the angular coefficients were obtained to determine the inhibition (*I*%) of the enzymatic activities by the equation:$$I\%=100\times \frac{\left(A-a\right)-\left(B-b\right)}{\left(A-a\right)}$$
where *A* is the angular coefficient of the straight-line equation (enzyme + substrate), *a* is the angular coefficient of the equation of the line (substrate), *B* is the angular coefficient of the straight-line equation (enzyme + substrate + sample), and *b* is the value of the angular coefficient of the straight-line equation (enzyme + sample).

The 50% inhibitory concentrations (IC_50_) were determined through “response versus concentration” plots using the linear least-squares regression model.

#### 2.3.5. Determination of Kinetic Parameters

Kinetic parameters were determined using the same experimental conditions as described above for each enzyme [21]. The reactions were prepared using increased substrate concentrations (16 to 0.01042 mM), both in the absence and presence of PPC80, PPC82, PPC84, PPC89, and PPC101 derivatives or positive control (orlistat or acarbose). The enzyme concentrations were maintained as described above. The absorbance of the products was measured at 405 nm using a microplate reader (Thermoplate^®^, TP-Reader, Wuxi, China) as a function of time (60 s). The absorbance values were converted into product concentration (µmol/L) using standard curves of glucose (α-amylase) and *p*-nitrophenol (pancreatic lipase and α-glucosidase). The value of the initial velocity (*v*_0_) of enzymatic reactions was estimated to create the “*v*_0_ versus substrate concentration” graph. Kinetic constants (*K_m_* and *V_max_*) and slope were calculated, and the inhibition model was verified using Lineweaver–Burk plots [21].

### 2.4. Statistical Analysis

The data were subjected to the analysis of variance (ANOVA) and Tukey’s test (*p* < 0.05) to determine the differences between mean groups using the GraphPad Prism 5 program. Data were presented as mean ± S.E.M.

## 3. Results

### 3.1. Synthesis of Protected Amino Acid Derivatives

The synthesis started by reacting Boc-protected L-isoleucine amino acid with EDC.HCl as carboxylic acid coupling. After 30 min, the vessel was charged with the corresponding nucleophile in the presence of racemic camphorsulphonic acid (+/−)-CSA as an organocatalyst [17]. The corresponding synthetic amino acid derivatives, PPC80, PPC82, PPC84, PPC89, and PPC101, were attaining in yields ranging from 67 to 80% (Figure 1). It is worth to mentioning that no epimerization process was observed. The characterization data are in agreement with those previously described in the literature [17]. The products (PPC80, PPC82, PPC84, PPC89, and PPC101) were then used to carry out inhibitory activity assays against digestive enzymes.

### 3.2. Inhibitory Effect of Amino Acid Derivatives on Pancreatic Lipase

The results showed that the inhibitory effects of PPC82, PPC80, and PPC84 amino acid derivatives on pancreatic lipase activity were concentration-dependent (Figure 2). PPC80 (584 µM) was more active (*p* < 0.05) than orlistat (807 µM, 60% inhibition) at a lower concentration, showing an inhibitory effect of about 65% on pancreatic lipase activity (Figure 2A). However, PPC82 (477 µM) was more effective in inhibiting pancreatic lipase at a lower concentration than orlistat (807 µM), with a response of about 86% (Figure 2B). Moreover, PPC84 (1392 µM) exerted a similar inhibitory effect on pancreatic lipase at a higher concentration than orlistat (807 µM), showing an inhibitory effect of about 62% (Figure 2C). In this assay, PPC89 and PPC101 did not show inhibitory action on the reference enzyme.

As shown in Table 1, the IC_50_ values were also determined. PPC80 and PPC82 showed better IC_50_ values than orlistat at the lower concentrations of 475.30 ± 8.25, 167.00 ± 6.25, and 587.70 ± 14.90 µM, respectively (*p* < 0.05). On the contrary, PPC84 (1023.00 ± 20.34 µM) had a higher IC_50_ value at a lower concentration than orlistat (*p* < 0.05), thereby showing a lower inhibitory effect on pancreatic lipase activity.

### 3.3. Kinetic Parameters on Pancreatic Lipase Activity

To determine the inhibitory mechanism against pancreatic lipase, PPC80, PPC82, and PPC84 were evaluated for kinetic parameters (*K_m_*, *V_max_*, and Slope) and Lineweaver–Burk profiles (Table 2 and Figure 3). In the “no inhibitor” group, *K_m_* and *V_max_* values were 0.19 ± 0.006 mM and *V_max_* of 68.65 ± 0.41 µM/min, respectively. Orlistat (101 µM), the reference drug, was able to reduce the reaction velocity with *V_max_* equal to 68.34 ± 0.40 µM/min and *K_m_* of 0.14 ± 0.001 mM. The addition of 146 µM PPC80 (*V_max_* = 68.82 ± 0.57 µM/min and *K_m_* = 0.16 ± 0.004 mM), 159 µM PPC82 (*V_max_* = 69.13 ± 0.57 µM/min and *K_m_* = 0.20 ± 0.004 mM), and 174 µM PPC84 (*V_max_* = 62.76 ± 0.3 µM/min and *K_m_* = 0.10 ± 0.003 mM) decreased the enzyme-substrate reaction rate (Table 2). In addition, the slope values increased in the presence of these compounds, showing that the reaction was slower, as confirmed by the Lineweaver–Burk plots (Figure 3). Further observing the data in Table 2, PPC80 and PPC82 produced *V_max_* statistically equal to the “no inhibitor” group and different *K_m_* values (*p* < 0.05), which is indicative of a competitive mechanism. However, as it produces different *V_max_* and *K_m_* values in relation to the “no inhibitor” group, PPC84 must follow a mixed or non-competitive inhibition mechanism.

### 3.4. Inhibitory Effect of Amino Acid Derivatives on Pancreatic α-Amylase

The results showed that the inhibitory effects of PPC80, PPC82, PPC84, PPC89, and PPC101 amino acid derivatives on pancreatic α-amylase activity were concentration-dependent (Figure 4). PPC80 (730 µM), PPC82 (1590 µM), PPC84 (870 µM), PPC89 (675 µM), and PPC101 (532 µM) inhibited pancreatic α-amylase by nearly 93%, 94%, 74%, 90%, and 86% (*p* < 0.05), respectively, while acarbose (620 µM) reduced the specific enzymatic activity by about 86% (Figure 4A–E). PPC101 was more active than acarbose at a lower concentration, while PPC80 and PPC82 showed better inhibitory effects than acarbose at higher concentrations. Moreover, PPC89 (405 µM) produced the same inhibitory effect as acarbose (620 µM).

The IC_50_ values showed the inhibitory potential of PPC80, PPC82, PPC84, PPC89, and PPC1010 derivatives (Table 1). PPC89 (171.30 ± 13.57 µM) and PPC101 (162.00 ± 1.73 µM) had lower IC_50_ values than acarbose (326.00 ± 3.21 µM), thereby showing a more potent inhibitory activity against pancreatic amylase, while PPC82 and PPC84 were less effective in inhibiting the activity of the enzyme (*p* < 0.05). Moreover, PPC80 showed a similar suppressive potential as the reference compound (acarbose). The inhibitory effects were corroborated in Figure 4, where the inhibition of the pancreatic α-amylase enzyme occurred at higher concentrations of PPC82 and PPC84.

### 3.5. Kinetic Parameters against Pancreatic α-Amylase

The kinetic parameters of pancreatic α-amylase activity for the PPC89, PPC101, PPC80, PPC84, and PPPC82 amino acid derivatives were determined (Table 3). In the absence of enzyme inhibitors, the reaction had a *K_m_* of 0.06 ± 0.006 mM, while *V_max_* was 100.70 ± 0.34 µM/min). By inhibiting pancreatic amylase, acarbose and amino acid derivatives were able to reduce the reaction rate (Table 3). PPC82, PPC89, and PPC101 produced *V_max_* equal to the “no inhibitor” group with *K_m_* different from this group (*p* < 0.05), suggesting that these compounds present a competitive-type inhibition mechanism. On the contrary, the *V_max_* values of PPC80 and PPC84 were different from the “no inhibitor” group, which indicates that these derivatives follow another inhibitory mechanism. However, as observed, the slope values increased in the presence of these compounds, showing that the reaction was slower, which was confirmed by the Lineweaver–Burk plots (Figure 3). Furthermore, for the compounds with competitive inhibition, the increase in the substrate concentration caused an increase in the slope, as observed in Table 2 and Figure 5.

### 3.6. Inhibitory Effect of Amino Acid Derivatives on α-Glucosidase

As shown in Figure 6, the inhibitory effects of PPC84, PPC89, and PPC101 amino acid derivatives on α-glucosidase activity were concentration-dependent. PPC84 (435 µM), PPC89 (674 µM), and PPC101 (67 µM) inhibited α-glucosidase by nearly 66, 78, and 64% (*p* < 0.05), respectively, inhibiting enzyme activity at lower concentrations than acarbose (positive control). These derivatives were also more effective at higher concentrations (Figure 6A–C). Moreover, PPC80 and PPC82 did not show inhibitory action against the tested enzyme.

The IC_50_ values showed the inhibitory potential of PPC84, PPC89, and PPC101 amino acid derivatives on α-glucosidase activity (Table 1). PC89 (353.00 ± 6.03 µM), PPC84 (321.30 ± 2.03 µM), and PPC101 (51.00 ± 1.73 µM) derivatives had lower IC_50_ and therefore were more effective for inhibiting α-glucosidase than acarbose (IC_50_ = 639.00 ± 4.62 µM). It is worth noting that PPC101 was 12-fold more potent than acarbose, exhibiting an outstanding potential to inhibit the target enzyme.

### 3.7. Kinetic Parameters of α-Glucosidase

Kinetic parameters were also evaluated against α-glucosidase (Table 4). The enzyme–substrate reaction of the “no inhibitor” group (*K_m_* = 0.183 ± 0.009 mM and *V_max_* = 62.42 ± 1.14 μM/min) was faster than acarbose (*K_m_* = 0.106 ± 0.003 mM and *V_max_* = 63.35 ± 1.43 µM/min). The *V_max_* values of PPC84 (62.90 ± 0.60 µM/min), PPC89 (62.30 ± 1.33 µM/min), and PPC101 (63.09 ± 1.45 µM/min) were statistically equal to the reference group (no inhibitor) (*p* < 0.05), while the *K_m_* values were different in this group. These data and the Lineweaver–Burk plots (Figure 7) show that amino acid derivatives follow a competitive-type inhibition mechanism, since *V_max_* values are maintained during enzymatic reactions. Furthermore, slope values ranged from 2.93 to 4.94 min^−1^ and increased with increasing substrate concentration, which is also a feature of competitive inhibitors.

## 4. Discussion

The structure of amino acid derivatives suggested that the hydrocarbon chain is involved in their inhibitory effects, since compounds with a side chain with more than eight carbon atoms did not inhibit pancreatic lipase. The inclusion of an amino group at the carbon side chain of PPC84 (Figure 1) may have led to additional hydrogen bonding interactions (non-covalent interactions) at the catalytic site, resulting in an impaired ability to competitively inhibit the enzymatic activity. However, the possibility of the amine to act as a nucleophile (covalent bonding) cannot be ruled out [22]. Among these compounds, PPC82 (six carbons in the side chain) was more potent, confirming the inhibition data, while PPC84 was less active in inhibiting pancreatic lipase (Figure 3).

Considering the kinetic parameters, as they presented *V_max_* equal to the “no inhibitor” group, PPC80 and PPC82 were defined as competitive inhibitors. In this type of inhibition, the slope increased with increasing substrate concentration [S], which is observed in the data in Table 2. In contrast, PPC84 produced a different *V_max_* than the “no inhibitor” group with reduced slope and *K_m_*, which may be related to the interaction of the side chain amino group with another enzymatic site producing a non-competitive or mixed type of inhibition [21]. This type of inhibition can also be seen on the Lineweaver–Burk plot as an increased ordinate intercept with no effect on the abscissa intercept (−1/*K_m_*) (Figure 3) [23].

These findings are consistent with those of previous studies assessing the effects of amino acid and peptide derivatives on pancreatic lipase activity. Ngoh and Gan [24] identified different peptides from the common bean (*Phaseolus vulgaris*) that inhibited pancreatic lipase in the range of 23–87%. Polylysine is a synthetic peptide that also acts as a lipase inhibitor, showing a remarkable inhibition (80%) on the activity of porcine pancreatic lipase at a concentration of 100 mg/mL [25]. Furthermore, synthetic peptides [26] and hydrolyzed peptides [22] inhibited pancreatic lipases with IC_50_ values below 50 µM.

The results of the present study showed that PPC80 and PPC82 have inhibitory potential on the activity of pancreatic lipase, which may promote a reduction in intestinal fat absorption and potentially affect body weight [27]. Therefore, PPC80 and PPC82 derivatives are promising therapeutic agents for the treatment of obesity and lipid disorders, since orlistat (an anti-obesity drug) is associated with nephrotoxicity, hepatotoxicity, and gastrointestinal side effects [28].

The function of amino acid derivatives may be associated with the size of the hydrocarbon chain, since compounds such as PPC89 and PPC101, which exhibit a high number of carbon atoms after nitrogen in their aliphatic chains, were more effective in inhibiting the activity of pancreatic α-amylase at low IC_50_ values (Table 1). PPC80 (eight carbons) showed a similar inhibitory effect as acarbose, thus being the third most effective compound (Table 1). Moreover, PPC82 (six carbons) and PPC84 (three carbons and one amino group) showed the lowest inhibitory activities.

The structure–activity relationship may also be related to the inhibitory mechanism on pancreatic α-amylase, since PPC82, PP89, and PPC101 showed *V_max_* equal to the “no inhibitor” group with lower slope values (Table 3), which may involve a competitive type of inhibition. The *V_max_* values of PPC80 and PPC84, with eight and ten carbons in the side chain, respectively, differed from the “no inhibitor” group, showing that these compounds have a non-competitive or mixed inhibition mechanism, that is, they do not act on the same substrate site [21]. As they are considered competitive, the slopes of PPC82, PPC89, and PPC101 increased with the increase in [S] but produced *K_m_* values different from the “no inhibitor” group, which can be confirmed through the Lineweaver–Burk plot (Figure 5).

The catalytic mechanism of the α-amylase family is stable and specific because of the α-retaining double-displacement reaction. This two-step mechanism is a distinctive feature of the α-amylase family and may contribute to its broad specificity due to the attachment of different domains to the catalytic site or to extra sugar-binding subsites around the catalytic site [29]. However, the carboxylic groups of aspartate and glutamate residues can act as acid/base catalysts and nucleophilic reagents during the formation of covalent intermediates in the catalytic cycle. The presence of chloride anions may lead to activation and facilitate the protonation of a carboxyl group [30].

An increasing number of studies have shown that synthetic compounds derived from amino acids and peptides exhibit an inhibitory action on α-amylase [31,32,33]. Two α-amylase inhibitor peptides (GGSK and ELS) were obtained from red seaweed (*Porphyra* species) with IC_50_ values of 2.58 ± 0.08 and 2.62 ± 0.05 mM for GGSK and ELS, respectively [31]. Another study reported peptides extracted from basil (*Ocimum basilicum*) seeds that showed 36% inhibition on α-amylase [33]. Similarly, González-Montoya et al. [32] also identified peptides from soy (*Glycine max*) protein capable of inhibiting pancreatic α-amylase activity at IC_50_ values ranging from 0.16 to 8.30 mg/mL.

The increase in the side chain and the inclusion of the amino group allowed a greater inhibitory action on α-glucosidase, as PPC101 (IC_50_ = 51.00 ± 1.73 µM) was about 12-fold more potent than acarbose (IC_50_ = 639.00 ± 4.62 µM) against this enzyme (Table 2). Although with less inhibitory action, PPC84 and PPC89 were more active than the reference drug (acarbose), indicating that the molecular structure of these compounds influenced the α-glucosidase activity. In addition, based on the kinetic parameters, these derivatives showed *V_max_* values equal to the “no inhibitor” group, which characterized a competitive-type mechanism between the substrate and the compounds occupying the same enzymatic site [21]. This type of inhibition has a different *K_m_*, and the slope increases with increasing substrate.

Pancreatic α-amylase and α-glucosidase are critical enzymes involved in the digestion of dietary starch, catalyzing the release of oligosaccharides that are further degraded into glucose. Therapeutic approaches for the treatment of type two diabetes include the inhibition of these enzymes to decrease the absorption of glucose in the digestive tract and reduce postprandial hyperglycemia [34,35]. Acarbose, miglitol, and voglibose are major inhibitors that reduce the rate of glucose absorption, attenuating the postprandial increase in plasma glucose levels, and thus helping in the treatment of obesity [36,37]. Our results indicate that amino acid derivatives are potent inhibitors of pancreatic α-amylase and can be promising agents for the treatment of diabetes and metabolic disorders.

Several studies have reported the promising potential of amino acid and peptide derivatives as α-glucosidase inhibitors [37,38,39]. For example, KLPGF and NVLQPS peptides obtained from albumin showed inhibitory activity on α-glucosidase at IC_50_ values of 59.5 ± 5.7 µM and 100.0 ± 5.7 μM, respectively [39]. In this study, the inhibitory activity of the KLPGF peptide motif was similar to that of acarbose (IC_50_ = 60.8 μM). Furthermore, three peptides isolated from quinoa (*Chenopodium quinoa*) showed similar inhibitory activities against α-glucosidase [37]. Singh and Kaur [38] reported serine-threonine-tyrosine-valine-containing peptides isolated from the endophytic fungi *Acacia nilotica* that exhibited potent inhibitory effects against α-glucosidase at low IC_50_ values (3.75 µg/mL).

The human α-glucosidase is an enzyme found in the epithelium of the small intestine that catalyzes starch breakdown and the consequent release of glucose. Therefore, inhibition of this enzyme constitutes a promising strategy for reducing serum glucose levels in metabolic diseases, including type two diabetes [20]. Our results showed that PPC89, PPC84, and PPC101 amino acid derivatives inhibit α-glucosidase, exhibiting potential as agents for lowering blood glucose levels in carbohydrate-related metabolic diseases.

## 5. Conclusions

In summary, the results showed that PPC80, PPC82, PPC84, PPC89, and PPC101 amino acid derivatives are potential inhibitors of lipase, α-amylase, and α-glucosidase enzymes. For instance, PPC80, PPC82, and PPC84 inhibited pancreatic lipase with IC_50_ values as low as 167 µM via competitive or mixed mechanisms. The activity of pancreatic α-amylase was suppressed by PPC80, PPC82, PPC84, PPC89, and PPC101, with IC_50_ values in a range of 162–519 µM, which acted as competitive or mixed inhibitors. In addition, PPC84, PPC89, and PPC101 also presented an inhibitory effect on α-glucosidase, with IC_50_ values as low as 51 µM acting as competitive inhibitors. The present study supports that amino acid derivatives are promising therapeutic agents for metabolic disorders, including type II diabetes and obesity. However, further pharmacological and toxicological investigations are needed to ensure their safe use as medicines.

## Figures and Tables

**Figure 1 biomolecules-13-00953-f001:**
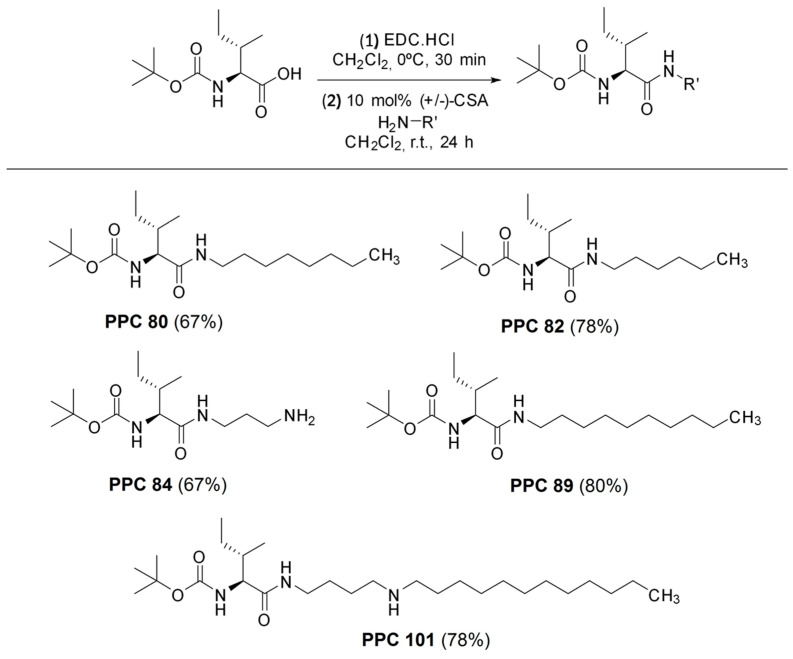
Synthesis and molecular structure of amino acid derivatives.

**Figure 2 biomolecules-13-00953-f002:**
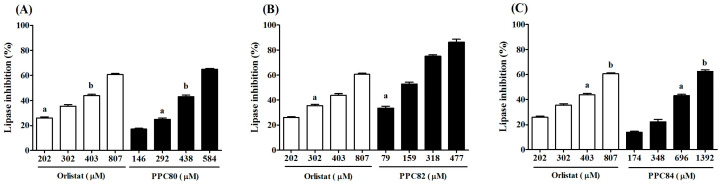
Inhibitory effects of amino acid derivatives on pancreatic lipase activity. Each bar represents the mean ± S.E.M. (*n* = 3). (**A**) PPC80. (**B**) PPC82. (**C**) PPC84. Repeated letters in the same figure indicate that group means did not show statistically significant differences after ANOVA and Tukey’s test (*p* < 0.05).

**Figure 3 biomolecules-13-00953-f003:**
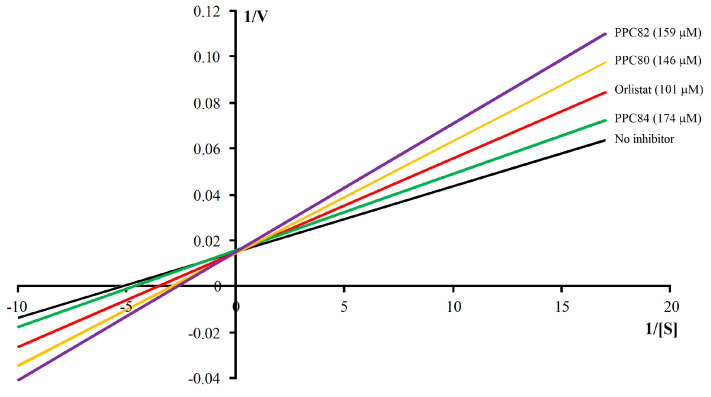
Lineweaver-Burk kinetic profile of amino acid derivatives against pancreatic lipase.

**Figure 4 biomolecules-13-00953-f004:**
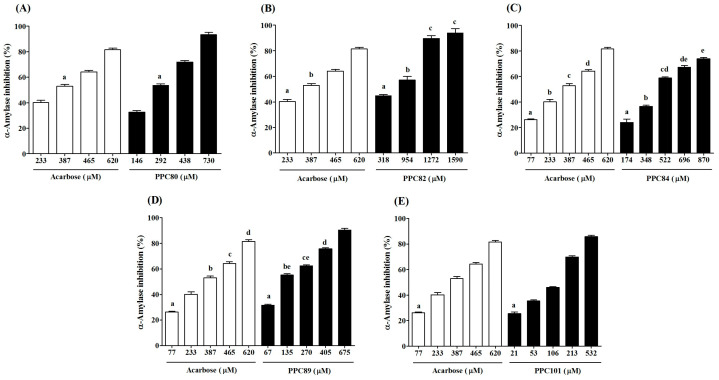
Inhibitory effect of amino acid derivatives on pancreatic α-amylase. Each bar represents the mean ± S.E.M. (*n* = 3). (**A**) PPC80. (**B**) PPC82. (**C**) PPC84. (**D**) PPC89. (**E**) PPC101. Repeated letters in the same figure indicate that group means did not show statistically significant differences after ANOVA and Tukey’s test (*p* < 0.05).

**Figure 5 biomolecules-13-00953-f005:**
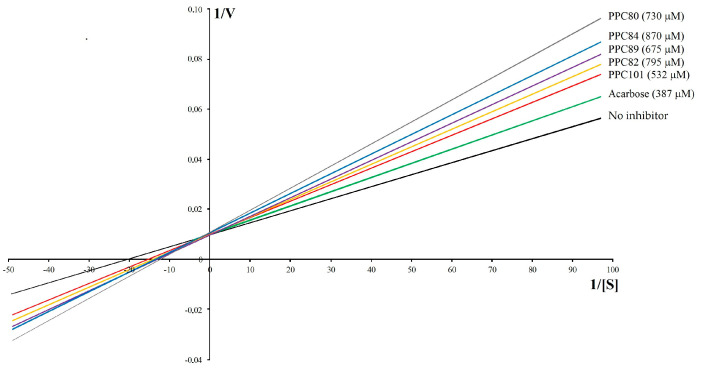
Lineweaver-Burk kinetic profile of amino acid derivatives against pancreatic amylase.

**Figure 6 biomolecules-13-00953-f006:**
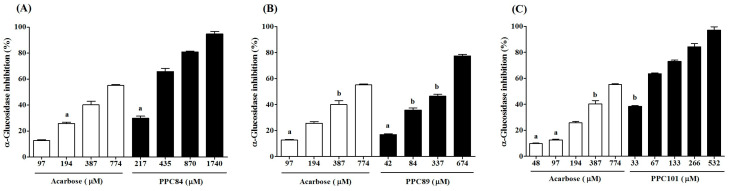
Inhibitory effect of amino acid derivatives on α-glucosidase. Each bar represents mean ± S.E.M. (*n* = 3). (**A**) PPC84. (**B**) PPC89. (**C**) PPC101. Repeated letters in the same figure indicate that group means did not show statistically significant differences after ANOVA and Tukey’s test (*p* < 0.05).

**Figure 7 biomolecules-13-00953-f007:**
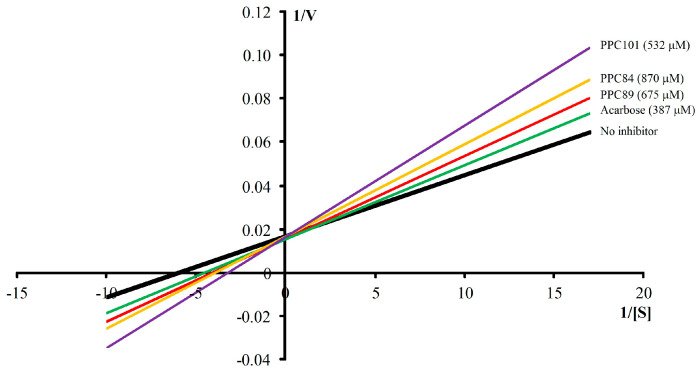
Lineweaver-Burk kinetic profile of amino acid derivatives against α-glucosidase.

**Table 1 biomolecules-13-00953-t001:** IC_50_ values of amino acid derivatives against pancreatic lipase, pancreatic α-amylase, and α-glucosidase.

Group	IC_50_ (µM)		
	Pancreatic Lipase	Pancreatic α-Amylase	α-Glucosidase
Orlistat	587.70 ± 14.90	-	-
Acarbose	-	326.00 ± 3.21 ^a^	639.00 ± 4.62
PPC80	475.30 ± 8.25	275.70 ± 5.21 ^a^	-
PPC82	167.00 ± 6.25	519.00 ± 19.97 ^b^	-
PPC84	1023.00 ± 20.34	493.00 ± 10.97 ^b^	321.30 ± 2.03
PPC89	-	171.30 ± 13.57 ^c^	353.00 ± 6.03
PPC101	-	162.00 ± 1.73 ^c^	51.00 ± 1.73

Each value represents the mean ± S.E.M (*n* = 3). Repeated letters (a–c superscript) in the same column indicate that group means did not show statistically significant differences after ANOVA and Tukey’s test (*p* < 0.05).

**Table 2 biomolecules-13-00953-t002:** Kinetic parameters of PPC80, PPC82, and PPC84 against pancreatic lipase.

Group	Concentration (µM)	*K_m_* (mM)	*V_max_* (µM/min)	Slope (min^−1^)
No inhibitor	-	0.19 ± 0.006 ^a^	68.65 ± 0.41 ^a^	2.77
Orlistat	101	0.14 ± 0.001	68.34 ± 0.40 ^a^	4.10
PPC80	146	0.16 ± 0.004	68.82 ± 0.57 ^a^	4.65
PPC82	159	0.20 ± 0.004 ^a^	69.13 ± 0.57 ^a^	5.79
PPC84	174	0.10 ± 0.003	62.76 ± 0.35	3.19

Each value represents the mean ± S.E.M (*n* = 3). Letter “a” superscript in the same column, means did not show statistically significant differences in relation to the “no inhibitor” group after ANOVA and Tukey’s test (*p* < 0.05).

**Table 3 biomolecules-13-00953-t003:** Kinetic parameters of amino acid derivatives against pancreatic α-amylase.

Group	Concentration (µM)	*K_m_* (mM)	*V_max_* (µM/min)	Slope (min^−1^)
No inhibitor	-	0.060 ± 0.006	100.70 ± 0.34 ^a^	0.59
Acarbose	387	0.033 ± 0.002 ^a^	100.30 ± 0.89 ^a^	0.66
PPC80	730	0.041 ± 0.001 ^a^	91.75 ± 0.48	0.89
PPC82	795	0.036 ± 0.001 ^a^	99.70 ± 1.20 ^a^	0.72
PPC84	870	0.038 ± 0.003 ^a^	94.35 ± 0.51	0.80
PPC89	675	0.038 ± 0.003 ^a^	99.73 ± 1.77 ^a^	0.76
PPC101	532	0.035 ± 0.002 ^a^	100.00 ± 1.55 ^a^	0.70

Each value represents the mean ± S.E.M (*n* = 3). Letter “a” superscript in the same column, means did not show statistically significant differences in relation to the “no inhibitor” group after ANOVA and Tukey’s test (*p* < 0.05).

**Table 4 biomolecules-13-00953-t004:** Kinetic parameters of amino acid derivatives against α-glucosidase.

Group	Concentration (µM)	*K_m_* (mM)	*V_max_* (µM/min)	Slope (min^−1^)
No inhibitor	-	0.183 ± 0.009	62.42 ± 1.14 ^a^	2.93
Acarbose	387	0.106 ± 0.003	63.35 ± 1.43 ^a^	3.35
PPC84	870	0.135 ± 0.007	62.90 ± 0.60 ^a^	4.29
PPC89	675	0.128 ± 0.001	62.30 ± 1.33 ^a^	4.11
PPC101	532	0.156 ± 0.002	63.09 ± 1.45 ^a^	4.94

Each value represents the mean ± S.E.M (*n* = 3). Letter “a” superscript in the same column, means did not show statistically significant differences in relation to the “no inhibitor” group after ANOVA and Tukey’s test (*p* < 0.05).

## Data Availability

The data presented in this study are available from the authors upon request.

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
