# Peer review of "Inhibitory Potential of Synthetic Amino Acid Derivatives against Digestive Enzymes as Promising Hypoglycemic and Anti-Obesity Agents"

_biomolecules, 2023, doi:10.3390/biom13060953_

Round 1

Reviewer 1 Report

The paper entitled Inhibitory potential of synthetic amino acid derivatives against digestive enzymes-promising hypoglycemic and anti-obesity agents describes the effect of five N-Boc protected amino amides against three digestive enzymes, pancreatic lipase, pancreatic α-amylase and α-glucosidase.

The five synthetic compounds here studies were previously described by some of these authors in an interesting paper published five years ago.

The proposed research work is inspiring and fits the scope of the journal. However, in my opinion, the paper is not well solved, with a lack of rigour in the description of the results and even in the discussion part. Both parts are repetitive and unattractive.

Suggestions and comments

  1. Check and modify the title the terms amino acid derivatives are a little bit forced
  2. Why don't the five described compounds have consecutive numbers?
  3. Express IC50 values with decimal places, eliminating the powers of 10. For instance, PPC80, IC50 = 0.475±0.08 µM or as appropriate.
  4. Check and modify expressions like “PPC84 (10.23 ± 0.20 x 10-1 mmol/L) had a higher IC50 value at a lower concentration than orlistat” because the IC50 value of a compound is not concentration dependent. There are many similar comments to this throughout the manuscript
  5. The meaning of the superscripts a-c is not indicated in the tables
  6. Figure 3 C. The compound PPC84 does not appear to behave as a competitive inhibitor of pancreatic lipase. The Vmax value decreases (also see Table 2).
  7. The data included in Tables 2-4 are confusing regarding the Km values. In addition, the terms Km and Ki are used interchangeably in the text. Check and correct this. It would be interesting to also include the slope value.
  8. The discussion is not very focused on the results obtained. It looks like an introduction to the topic.

Reviewer 2 Report

The paper of F. Campos da Silva et al describes a typical research approach in vitro, using newly synthesized (and original) compounds to attempt to 1/ inhibit a key enzyme – lipase – and 2/ build the basis for future work(s) to turn those compounds into actual, marketed drug(s).

The approach is interesting and well described.

I only found minor points to ameliorate the present paper, and a coupled of suggestion(s) for future development of the work.

A naïve reader ( = me) could ask if those compounds have not, potentially, detergent-like property. This should be verify, with simple tests – I believe with blood red cells. To see if those molecules (at those concentrations) are not disturbing the membrane architecture.

A very early approach of specificity is necessary. This is not trivial and should be done in order to avoid future deception when the lack of such specificity is discovered. Prepare the reader to this future, at least by mentioning future experiments.

Line 339: did the Authors attempt to dock the compound(s) into any of the enzymes structures?

Major: it is not clear if the amino acids were used with or without their protection? Please confirm that the structures of the Figure 1 are those of the actual compounds used in the assay(s).

Major: Lines 329: Please, mention the limitation of orlistat in therapeutics. I believe they were voices claiming the intestinal hemorrhagic risk of this compound, no?

Minor: Line 86 Please, avoid the use of the term “drug”, unless the compound is available on the market. Prefer the use of “compound” or ”agent”. Except for orlistat, of course.

Minor: Line 101 and all the Tables why using “10−1 mmol/L” instead of mM??? As well as in the conclusion paragraph (and elsewhere?)

Minor: Figure 1: please present the structure of orlistat, even if this is widely known.

Minor: It seems to me that the Ki could have and have been calculated. Why not summarizing those parameters on a table?

Minor: line 390 Attention! The Ki should be expressed in M, not in ng/mL throughout the paper.

Round 2

Reviewer 1 Report

The paper has been checked and modified by the authors. They have made appreciable changes to improve the clarity and quality of the manuscript. However, in my opinion, there are still some modifications and corrections that should be performed.

  1. Figure 1. In order to clarify this scheme, please, replace the notations (A)-(E) with the compound codes PPC80-PPC101 and eliminate the orlistat structure. If the authors want to include in the manuscript the reference drugs structure it should be done in an additional figure.
  2. On page 8, line 255, there is a mistake. Replace PPC89 with PPC84, please.
  3. On page 12, line 356, replace “Cl ion” with “chloride anion”.

The results and discussion part should be checked again. Some paragraphs are unexciting due to their repetitive content, for instance “As noted in Figure 2C, PPC84 concentration was almost 2-fold higher than the reference compound, which may be associated with its high concentration (1392 µM) observed in the inhibitory effect assay (Figure 2C) and shows a low inhibitory affinity with the target enzyme” (page 5, lines 193-196).

In addition, some phrases or expressions are incorrect or difficult to understand, like “In the absence of enzyme inhibitors, the reaction was faster than acarbose and had a Km of 0.183 ± 0.009 mM and a Vmax of 62.42 ± 1.14 μM/min (Table 4)” (page 9, lines 286-287) or “These data can also be confirmed in the Lineweaver-Burk plot whose intersection of PPC84 with the "no inhibitor" group is outside the y-axis (Figure 3)” (page 11, line 319-320).

Author Response

Dear Reviewer,

We appreciate the suggestions and comments, and we perform the review as requested.

Comment 1

  1. Figure 1. In order to clarify this scheme, please, replace the notations (A)-(E) with the compound codes PPC80-PPC101 and eliminate the orlistat structure. If the authors want to include in the manuscript the reference drugs structure it should be done in an additional figure.

Response:

We have replaced the notations and eliminated the structure of orlistat.

Comment 2

  1. On page 8, line 255, there is a mistake. Replace PPC89 with PPC84, please.

Response:

We have made the correction.

Comment 3

  1. On page 12, line 356, replace “Cl ion” with “chloride anion”.

Response:

We have made the replacement.

Comment 4

The results and discussion part should be checked again. Some paragraphs are unexciting due to their repetitive content, for instance “As noted in Figure 2C, PPC84 concentration was almost 2-fold higher than the reference compound, which may be associated with its high concentration (1392 µM) observed in the inhibitory effect assay (Figure 2C) and shows a low inhibitory affinity with the target enzyme” (page 5, lines 193-196).

Response:

In fact, the phrase was confusing. For better understanding, we removed it from the text.

Comment 5

In addition, some phrases or expressions are incorrect or difficult to understand, like “In the absence of enzyme inhibitors, the reaction was faster than acarbose and had a Km of 0.183 ± 0.009 mM and a Vmax of 62.42 ± 1.14 μM/min (Table 4)” (page 9, lines 286-287) or “These data can also be confirmed in the Lineweaver-Burk plot whose intersection of PPC84 with the "no inhibitor" group is outside the y-axis (Figure 3)” (page 11, line 319-320).

Response:

We have improved the text.
